# A Unique Relative of Rotifer Birnavirus Isolated from Australian Mosquitoes

**DOI:** 10.3390/v12091056

**Published:** 2020-09-22

**Authors:** Caitlin A. O’Brien, Cassandra L. Pegg, Amanda S. Nouwens, Helle Bielefeldt-Ohmann, Bixing Huang, David Warrilow, Jessica J. Harrison, John Haniotis, Benjamin L. Schulz, Devina Paramitha, Agathe M. G. Colmant, Natalee D. Newton, Stephen L. Doggett, Daniel Watterson, Jody Hobson-Peters, Roy A. Hall

**Affiliations:** 1Australian Infectious Diseases Research Centre, School of Chemistry and Molecular Biosciences, The University of Queensland, Brisbane, QLD 4067, Australia; caitlin.obrien@uqconnect.edu.au (C.A.O.); h.bielefeldtohmann1@uq.edu.au (H.B.-O.); jessica.harrison@uqconnect.edu.au (J.J.H.); b.schulz@uq.edu.au (B.L.S.); devina.paramitha@uq.net.au (D.P.); agathe.colmant@uq.net.au (A.M.G.C.); natalee.newton@uq.net.au (N.D.N.); d.watterson@uq.edu.au (D.W.); j.peters2@uq.edu.au (J.H.-P.); 2School of Chemistry and Molecular Biosciences, The University of Queensland, Brisbane, QLD 4067, Australia; c.pegg1@uq.edu.au (C.L.P.); a.nouwens@uq.edu.au (A.S.N.); 3Public Health Virology, Queensland Health Forensic and Scientific Services, Brisbane, QLD 4108, Australia; Ben.Huang@health.qld.gov.au (B.H.); David.Warrilow@health.qld.gov.au (D.W.); 4New South Wales Health Pathology, Westmead Hospital, Sydney, NSW 2145, Australia; John.Haniotis@health.nsw.gov.au (J.H.); Stephen.Doggett@health.nsw.gov.au (S.L.D.)

**Keywords:** birnavirus, mosquitoes, insect-specific virus, Aedes birnavirus, *Aedes notosrciptus*

## Abstract

The family *Birnaviridae* are a group of non-enveloped double-stranded RNA viruses which infect poultry, aquatic animals and insects. This family includes agriculturally important pathogens of poultry and fish. Recently, next-generation sequencing technologies have identified closely related birnaviruses in *Culex, Aedes* and *Anopheles* mosquitoes. Using a broad-spectrum system based on detection of long double-stranded RNA, we have discovered and isolated a birnavirus from *Aedes notoscriptus* mosquitoes collected in northern New South Wales, Australia. Phylogenetic analysis of Aedes birnavirus (ABV) showed that it is related to Rotifer birnavirus, a pathogen of microscopic aquatic animals. In vitro cell infection assays revealed that while ABV can replicate in *Aedes*-derived cell lines, the virus does not replicate in vertebrate cells and displays only limited replication in *Culex*- and *Anopheles*-derived cells. A combination of SDS-PAGE and mass spectrometry analysis suggested that the ABV capsid precursor protein (pVP2) is larger than that of other birnaviruses and is partially resistant to trypsin digestion. Reactivity patterns of ABV-specific polyclonal and monoclonal antibodies indicate that the neutralizing epitopes of ABV are SDS sensitive. Our characterization shows that ABV displays a number of properties making it a unique member of the *Birnaviridae* and represents the first birnavirus to be isolated from Australian mosquitoes.

## 1. Introduction

The family *Birnaviridae* comprises a group of non-enveloped viruses with bi-segmented, double-stranded RNA (dsRNA) genomes. Birnaviruses are unique among dsRNA viruses as they appear to share structural and phylogenetic characteristics with positive-sense single-stranded RNA ( + ssRNA) viruses [1,2].

To date, four genera have been established based on virus host range—*Avibirnavirus*, which contains only the type species infectious bursal disease virus (IBDV); *Blosnavirus*, containing blotched snakehead virus (BSNV); *Aquabirnavirus*, encompassing infectious pancreatic necrosis virus (IPNV) and its relatives; and *Entomobirnavirus*, comprising type species *Drosophila X virus* (DXV) and related viruses from insects [3]. A number of divergent viruses remain unassigned to any genus within the family including Rotifer birnavirus (RBV), a pathogen of the rotifer *Branchionus plicatilis* [4] and Drosophila birnavirus (DBV) which was identified in small RNA sequencing from *Drosophila melanogaster,* but does not phylogenetically group with the entomobirnaviruses [5].

The birnavirus genome is made up of two segments. Segment A encodes a polyprotein which contains the capsid protein precursor pVP2, endopeptidase VP4 and ribonucleoprotein VP3. VP4 is responsible for autoproteolytic processing of the polypeptide to release the three individual proteins. In some birnaviruses, a second open reading frame on this segment gives rise to a small non-structural protein [6]. Segment B encodes VP1, the viral RNA-dependent RNA polymerase (RdRP) which possesses an unusual A-B-C to C-A-B catalytic motif rearrangement and lacks the characteristic GDD motif found in other viral RdRPs [2,7,8,9].

Birnavirus replication occurs in the cytoplasm of infected cells and the immature birnavirus particle is formed by the association of pVP2 with VP3, in complex with the viral genome and VP1 [10,11,12,13,14,15]. Virus maturation occurs via a succession of cleavage events at the C-terminus of pVP2, leading to production of icosahedral particles primarily consisting of the mature form of VP2 in a T = 13 symmetry [6,12,15]. The resulting pVP2 cleavage products remain associated with the mature virion [6].

In 2011, the isolation of a new birnavirus which exhibited enhanced replication in the presence of a virulent strain of dengue-2 virus was reported [16]. The origin of this virus, named Espirito Santo virus (ESV), was unclear but its relatedness to DXV and inability to replicate in vertebrate cells suggested that it was a member of the *Entomobirnavirus* genus. Mosquito X virus (MXV) and Culex Y virus (CYV), which share high genetic similarity to ESV, have since been identified in wild-caught *Culex pipiens* and *Anopheles sinensis* mosquitoes in the absence of dengue virus [17,18]. The related Culicine-associated Z virus (CAZV) was identified from metagenomic sequencing data of *Aedes* (*Ochlerotatus*) *caspius* and *Ae.* (*Oc.*) *detritus* mosquitos from France [19]. Most recently, the isolation of Port Bolivar virus (PTBV) from *Ae. sollicitans* mosquitoes in America was reported [20]. Phylogenetic analyses suggest that ESV and the mosquito-associated birnaviruses form a subclade, separate from the *Drosophila*-associated viruses within the *Entomobirnavirus* genus [3,19,20].

The advent of next-generation sequencing technologies has shed light on a previously overlooked group of viruses in diptera, the insect-specific viruses (ISVs) [21]. As their name suggests, these viruses lack the ability to replicate in vertebrate systems but are maintained in insect populations, presumably via a variety of mechanisms including vertical, and oral–fecal route transmission [21]. ISVs have become the focus of increasing research due to their potential as biological control mechanisms, safe vehicles for vaccine delivery, and their contribution to elucidating viral evolution and virus–host interactions [22,23,24,25]. We have undertaken a virus discovery project in order to broaden our understanding of the virome of Australian mosquitoes, using a broad-spectrum, high-throughput ELISA-based method. This method, the monoclonal antibodies to viral RNA intermediates in cells (MAVRIC) system, utilizes two antibodies which recognize long (> 30 bp) dsRNA molecules, a common component of RNA virus replication, in a sequence-independent manner [26] and has been highly successful [25,27,28,29,30,31,32,33]. Here, we report the discovery of a birnavirus from Australian mosquitoes using this system. This virus is phylogenetically distinct from the *Entomobirnavirus* genus and its growth is restricted to *Aedes*-derived cell culture systems.

## 2. Materials and Methods

### 2.1. Mosquito Collection, Processing and Virus Culture

Mosquitoes were collected in dry ice-baited traps at various sites in New South Wales (NSW), Australia, between March and April in 2013 and 2014 as part of routine arbovirus surveillance in the region. Individual mosquitoes were identified using morphological criteria [29] and pooled into groups of up to 25 insects. Pools were mechanically homogenized in cell culture media with glass beads and passed through a 0.2 μm filter before inoculating onto *Aedes albopictus* larval (C6/36) cells. Methods for mosquito collection from Western Australia and Darwin have been described previously [29,34].

For MAVRIC screening, C6/36 cells were seeded at 50–70% confluence in 96-well plates one day prior to inoculation and incubated at 28 °C. The following day, 50 μL/well of mosquito homogenate was added to four replicate wells for each mosquito pool. West Nile virus subtype kunjin was used as a positive control for dsRNA production and cells inoculated with media alone (mock infection) served as a negative control. After 7 days, supernatant was harvested from inoculated wells and plates were fixed with 150 μL/well formaldehyde fixative buffer (4% formaldehyde, 0.5% *v*/*v* Triton X-100 in phosphate-buffered saline (PBS; 140 mM NaCl, 2.7 mM KCl, 6 mM Na_2_HPO_4_, 0.9 mM KH_2_PO_4_) for 10 min at 4 °C. After removing fixative, plates were allowed to dry for at least two days before fixed-cell ELISA was performed.

Plates were incubated with 150 μL/well TENTC blocking buffer (2% *w*/*v* casein, 0.05% *v*/*v* Tween-20, 10 mM Tris, 0.2 M NaCl, 1 mM EDTA, pH 8.0) for 1 h at room temperature. A cocktail of anti-dsRNA monoclonal antibodies (mAbs) 3G1 and 2G4 (MAVRIC) was freshly prepared by diluting hybridoma supernatant to a previously determined optimal concentration in TENTC blocking buffer. After removing blocking solution, 50 μL of MAVRIC cocktail was added to each well and plates were incubated at 37 °C for 1 h. Plates were washed four times with phosphate-buffered saline with 0.05% *v*/*v* Tween-20 (PBS-T) and goat anti-mouse HRP (DAKO) diluted in TENTC blocking buffer was added at 50 μL/well. After another incubation at 37 °C for 1 h, plates were washed 6 times with PBS-T. Finally, plates were incubated with 100 μL/well of substrate buffer (1 mM 2,2′-Azino-bis(3-ethylbenzothiazoline-6-sulfonic acid) (ABTS), 2.5 mM H_2_O_2_, in a solution prepared by mixing 0.1 M citric acid with 0.2 M Na_2_HPO_4_ to give a pH 4.2) for 1 h at room temperature, protected from light. Absorbance was measured at 405 nm and samples with an average OD_405 nm_ equal to or greater than twice that of the average mock reading were considered positive. MAVRIC-positive samples were tested for the presence of LNV by fixed-cell ELISAs using anti-LNV mAb 6E6 with the method described above [28].

Aedes birnavirus (ABV) was isolated from two pools of 25 *Ae. notoscriptus* mosquitoes (pool numbers 177833 and 178287) collected in Ballina, NSW in 2013. Mosquito homogenates, which were identified as MAVRIC positive, were inoculated onto C6/36 cells and cultured for 7 days at 28 °C. These samples were passaged twice in 96-well plates before upscaling to a 24 well plate for the 3rd passage.

MAVRIC-positive samples were tested for the presence of known mosquito-borne viruses by RT-PCR with the primers listed in Appendix A. Briefly, RNA was extracted from supernatants of MAVRIC-positive cultures using the Machery-Nagel Nucleospin viral RNA isolation kit as per the manufacturer’s instructions. One-step RT-PCR was performed on extracted RNA using the ThermoFisher Superscript III one-step RT-PCR kit. The following cycling conditions were used for all primer sets: reverse transcription: 45 °C/30 min; PCR: one cycle 94 °C/2 min; 40 cycles 94 °C/30 s, 45 °C/30 s, 68 °C/1 min; followed by a final extension cycle of 68 °C/5 min.

### 2.2. Next Gneration Sequencing and Bioinformatic Analyses

A confluent T175 flask of C6/36 cells was inoculated with 200 µL of supernatant from a passage 3 stock of isolate 177833 and incubated for 7 days at 28 °C. Harvested supernatant was clarified by centrifugation at 3000 rpm for 15 min at 4 °C.

RNA was extracted from the clarified supernatant using the QIAamp Viral RNA extraction kit (Qiagen, Hilden, Germany) with carrier RNA omitted. Residual cell line DNA was removed using DNA Wipeout (Qiagen). First-strand cDNA synthesis was generated using the Protoscript II kit using the supplied random primer mix (New England Biolabs; Ipswich, MA, USA). Second-strand DNA synthesis was performed using the NEBNext mRNA Second-Strand Synthesis Module (New England Biolabs). cDNA libraries were constructed using the Nextera XT library kit (Illumina; Singapore, Singapore) with barcoding of individual samples. These reactions were performed according to the conditions recommended by the respective manufacturers. Library sequencing was performed by the Australian Genome Research Facility (AGRF) on a HiSeq Illumina sequencer (125 nt paired end reads). Sequence assembly was performed using the de novo assembly tool in Geneious v8 at low sensitivity using 125 bp read lengths. Two large contigs were generated with both returning significant matches to members of the *Birnaviridae* family by BLASTx.

RNA was extracted from a 1st passage stock of isolate 178287 using the Machery-Nagel Nucleospin viral RNA isolation kit as per the manufacturer’s instructions. Purified RNA was sent to the Australian Genome Research Facility (Brisbane, Queensland) for sequencing using the Illumina HiSeq2000 platform. De novo assembled sequences for 177833 were then used as a scaffold to map reads from next-generation sequencing data for 178287 in Geneious v8.4.

MAFFT alignment was performed on 23 birnavirus polyprotein and 18 RdRP amino acid sequences using the CIPRES gateway [35]. Amino acid alignments were used to construct maximum likelihood trees in MEGA 7.0.26 using the Jones–Taylor–Thornton model with gamma substitutions and no deletions. The corresponding nucleotide sequences for these proteins were aligned using the MUSCLE algorithm in Geneious Prime v2019.1.3. Phylogenetic analyses of nucleotide sequence alignments were performed in MEGA 7.0.26 using the general time-reversible (GTR) model with gamma distribution (5 categories ( + G, parameter = 1.8479)) and invariable variation for some sites ([ + I], 2.92% sites).

Alignment of the ABV polyprotein and VP1 sequences with those of IPNV (PP: NP_047196, VP1: NP_047197), IBDV (PP: ANY27027, VP1: ANY27028), DXV (PP: NP_690836, VP1: NP_690806), RBV (PP: CAX33877, VP1: CAX33878), ESV (PP: AEW87521, VP1: AEW87520) and BSNV (PP: CAD30689, VP1: CAD30691) was performed using the ClustalW 2.1 alignment tool in Geneious Prime v2019.1.3.

Protein molecular weight predictions were performed using the ExPASy Compute pI/Mw tool (https://web.expasy.org/compute_pi/).

### 2.3. Cell Culture

C6/36 (*Ae. albopictus*, RNAi-deficient) and Mos55 (*Anopheles gambiae*) cells were cultured in Roswell Park Memorial Institute 1640 (RPMI 1640) medium supplemented with 5% fetal bovine serum (FBS). Chao ball (*Culex tarsalis*) and RML-12 (*Ae. albopictus*, RNAi competent) cells were maintained in Leibovitz-15 (L-15) medium supplemented with 10% tryptose phosphate broth and 5% FBS. All insect cells were incubated at 28 °C. The vertebrate cell lines DF-1 (*Gallus gallus*, chicken embryo fibroblast), BSR (*Mesocricetus auratus*, baby hamster kidney), Vero (*Cercopithecus aethiops*, African green monkey kidney), A549 (*Homo sapiens*, human lung), SW13 (*Homo sapiens,* human adrenal gland/cortex), and IFNAR -/- MEFs (*Mus musculus*, mouse embryonic fibroblast, type 1 interferon receptor knockout) were maintained in Dulbecco’s Modified Eagle Medium (DMEM) supplemented with 5% FBS. Madin–Darby canine kidney (MDCK, *Canis familiaris*) cells were maintained in RPMI 1640 supplemented with 10% FBS. Viper spleen-derived VSW cells (*Daboia russelii*, Russel’s viper epithelial) were grown in Minimal Essential Media (MEM) with Hank’s salts, 0.35g/L sodium bicarbonate, 10% FBS and amphotericin B. A6 (*Xenopus laevis*, South African clawed toad kidney epithelial) cells were maintained in RPMI 1640 with 10% FBS and amphotericin B. All vertebrate cells were grown at 37 °C with 5% CO_2_ except for VSW and A6 cells which were grown at 28 °C without CO_2_. All media were supplemented with 50U/mL penicillin, 50 µg/mL streptomycin and 2 mM L-glutamine.

### 2.4. Virus Culture and Antigen Preparation

Large-scale stocks of ABV isolate 177833 were generated by inoculating a 50%-confluent T175 flask of C6/36 cells with passage 3 virus at a multiplicity of infection (MOI) of 0.1 and incubating at room temperature for 1 h with agitation. Inoculum was then removed and replaced with 18 mL RPMI 1640 with 2% FBS before incubating at 28 °C for 7 days. Virus supernatant was harvested and clarified at 3000 rpm at 4 °C for 15 min, the FBS content was then increased to 10% before storing at −80 °C in 0.5 mL aliquots.

ABV titers were determined with a modified TCID_50_ method [36] using MAVRIC, as described in [26]. Briefly, 10-fold dilutions of virus (undiluted to 1:10^8^) were added to C6/36 cells in 96 well plates and incubated for 7 days at 28 °C. Cells were fixed with 150 μL/well formaldehyde fixative buffer (4% formaldehyde, 0.5% *v*/*v* Triton X-100 in phosphate-buffered saline, PBS) for 10 min at 4 °C. After removing the fixative, plates were allowed to dry for at least 2 days before fixed-cell ELISA was performed using MAVRIC to test for viral replication as described in Section 2.1. Virus titers were calculated using the Reed–Muench guidelines [36].

Infected cells (approximately 10^7.3^ cells) from which virus stocks were generated were washed once with PBS and harvested by incubating with 3.5 mL NP-40 lysis buffer (1% NP-40 in 150 mM NaCl, 50 mM Tris-HCl [pH 7.3], 1:100 protease inhibitor cocktail (P8340, Sigma, St Louis, MO, USA)) for 30 min at 4 °C. Lysate was then clarified by centrifugation at 10,000 g for 10 min at 4 °C and stored at −80 °C. Lysate aliquots were boiled before use for 10 min at 90 °C to inactivate virus.

A preparation of ABV-infected cell culture supernatant with minimal FBS was generated using the protocol outlined above for production of virus stocks. However, after 5 dpi, the supernatant was removed and cells were washed 3 times with sterile PBS before the medium was replaced with 18 mL serum-free RPMI 1640. After another 48 h, the serum-free supernatant was harvested and clarified by centrifugation and concentrated using a high-molecular-weight (100 K) MWCO Spin-X UF concentrator column (Corning, Deeside, England).

### 2.5. ABV Purification and Analysis of Virions

For purification, ten T175 flasks of C6/36 cells were infected with ABV at a MOI of 0.1, as described above. After 7 days, supernatant was harvested and centrifuged at 3000 rpm at 4 °C for 15 min. Precipitation of virions was performed by mixing 2 parts clarified supernatant with 1 part 40% polyethylene glycol 6000 (PEG6000) solution at 4 °C overnight, followed by centrifugation at 12,000 rpm for 1 h at 4 °C using a Beckman Coulter JLA-16.250 rotor. Precipitated virus was then further purified through a 20% sucrose cushion in NTE buffer (10 mM Tris-Cl, 1 mM EDTA, 120 mM NaCl, pH 8), in a Beckman Coulter SW32 Ti rotor at 28,000 rpm for 2 h at 4 °C. The virus pellet was incubated in 500 μL PBS overnight at 4 °C before resuspending. Finally, the virion preparation was layered onto a discontinuous gradient of 1.23 and 1.37 g/mL cesium chloride (CsCl) in PBS and centrifuged in a Beckman Coulter SW60 Ti rotor at 38,000 rpm for 21 h at 15 °C. The resulting layer containing virus was extracted and buffer exchanged 3 times into PBS to remove residual CsCl.

Purified virions were prepared for transmission electron microscopy (TEM) on glow-discharged carbon/formvar-coated copper grids and negatively stained with 1% uranyl acetate in ultrapure water. All imaging was performed on a JEOL 1011 transmission electron microscope. The average diameter of virions was determined using the ImageJ measurement tool

The infectivity of purified virions was assessed by inoculating C6/36 cells grown in triplicate in 24-well plates with approximately 3 μL of purified virions diluted in 200 μL culture medium per well, or virus culture supernatant at a MOI of 0.1. Inoculum was incubated on cells for 1 h at room temperature with agitation. After 1 h inoculum was removed, and cells were washed 3 times with PBS before replacing with 1 mL RPMI 1640 supplemented with 2% FBS per well and incubating for 5 days at 28 °C. Virus titers were determined by the modified TCID_50_ method described in Section 2.4.

### 2.6. In Vitro Infectivity Assays

Approximately 10^5^ cells were seeded onto glass coverslips in 24-well plates one day prior to infection. Triplicate wells were either inoculated with media only (mock control) or media containing ABV isolate 177833 at a MOI of 0.1 for 1 h at room temperature with agitation. Inoculum was then removed, and monolayers were washed 3 times with PBS before replacing with 1 mL per well of appropriate media with reduced FBS content (2%). For DF-1, A6 and VSW cells, 1 mL of respective maintenance media described in Section 2.3 was used.

For the mosquito cell assay, all cells were incubated at 28 °C for 7 days while the vertebrate cell assay was performed at 37 °C (or 28 °C for VSW, A6 and C6/36 control cells) for 5 days. C6/36 cells were used as a control for infection in both assays. Supernatant was harvested 2 h post-infection (hpi) and both supernatant and coverslips were harvested 5 and 7 days post-infection (dpi) for the vertebrate and insect cells assay, respectively. Cell culture supernatants were assessed for virus by titrating back onto C6/36 cells, as described in Section 2.4.

### 2.7. Immunofluorescence Assays

For initial virus characterization and in vitro insect and vertebrate cell culture assays, cells grown on glass coverslips were washed once with PBS before fixing with 1 mL formaldehyde fixative buffer (described in Section 2.1) for 10 min at 4 °C. After 10 min, the fixative buffer was removed, and coverslips were allowed to dry at room temperature for at least 2 days. To investigate the effect of fixative method on MAVRIC immunolabeling, coverslips were washed once with PBS and fixed with formaldehyde fixative buffer, as described above, or by completely submerging the coverslip in ice-cold 100% acetone for 5 min and air drying at room temperature for 15 min before storing at −20 °C.

Coverslips were incubated with 900 µL TENTC blocking buffer for 1 h at room temperature. Primary antibody staining was performed by incubating with 300 µL of MAVRIC or ABV-specific hybridoma supernatant (see Section 2.8) diluted in TENTC blocking buffer for 1 h at 37 °C. Coverslips were washed 3x with PBS-T and stained with 150 µL secondary antibody (Alexa Fluor 488-conjugated goat anti-mouse (H + L) or Alexa Fluor 594-conjugated goat anti-mouse IgM (µ chain)) diluted 1:500 in TENTC blocking buffer for 1 h at 37 °C. Secondary antibody was removed and nuclear staining was performed with 150 µL Hoechst 33342 for 5 min at room temperature, protected from light. Coverslips were washed 3x with PBST, followed by one wash with PBS (without Tween-20) and mounted on glass microscope slides using ProLong Gold Antifade mountant (ThermoFisher, Carlsbad, CA, USA). All coverslips were imaged using a Zeiss LSM 510 META confocal microscope.

### 2.8. Generation and Characterization of Monoclonal Antibodies Raised to ABV

All animal procedures had received prior approval from The University of Queensland Animal Ethics Committee (AEC #SCMB/329/15/ARC) and, where necessary, were performed under ketamine:xylazine anaesthesia. Six-week-old BALB/c mice (Animal Resources Centre, Murdoch, Western Australia, Australia) were immunized twice two weeks apart via the subcutaneous route (s.c.) with purified ABV, along with inulin-based adjuvant Advax (Vaxine Ltd., Adelaide, Australia). Mice were kept on clean bedding and given food and water ad libitum. Immunized mice were bled via the tail vein at least two weeks following immunization and the sera tested for evidence of seroconversion to ABV using a fixed-cell ELISA as previously described [34]. Four days before fusions were performed, mice received a final boost via the intravenous route. For the first fusion, the mouse received a preparation of whole virus 4 months after the second s.c. injection. For the second fusion, the mouse received a preparation of purified virus which had been boiled at 90 °C for 10 min, 13 months after the second s.c. immunization. Spleens were harvested under sterile conditions and processed by passing through a 70 µm cell strainer with 10 mL RPMI. Fusion was induced between spleen cells and NS0 myeloma cells by the drop-wise addition of PEG-1500 (1g/mL in RPMI) at 37 °C. Fused cells were incubated overnight at 37 °C with 5% CO_2_ in a conditioned flask. Selection for hybridomas was performed by the addition of HAT supplement (hypoxanthine, aminopterin, thymidine; Hybri-Max, Sigma-Aldrich) and cells were plated in 96-well plates [37]. Hybridomas secreting ABV-reactive antibodies were identified by ELISA on ABV-infected C6/36 cell plates, as described in Section 2.1. However, initial screening was performed on plates fixed overnight at 4 °C with 200 µL/well acetone fixative buffer (20% *v*/*v* acetone, 0.02% *w*/*v* bovine serum albumin (BSA) in PBS).

Antibody isotyping was performed using Mouse Monoclonal Antibody Isotyping Reagents (Sigma-Aldrich, ISO2), as described previously [32].

Mouse immune serum and mAbs were tested against ABV-infected cell lysate by Western blot, as described previously [27,28]. Immune serum was also tested against purified virions, virus stock supernatant, and virus grown in minimal serum by Western blot.

Microneutralization assays were performed using 2-fold dilutions of hybridoma culture supernatant (1:2–1:256) or mouse immune serum (1:20–1:2480) and MAVRIC ELISA was used to measure neutralization effect, as described in [27].

### 2.9. Protein Analysis and Mass Spectrometry for Protein Identification

Purified virions were reduced with 80 mM DTT and boiled at 90 °C for 3 min before protein separation on precast 4–12% Bis-Tris SDS-PAGE gels (NuPAGE, ThermoFisher Scientific) at 175 volts for 45 min. Separated proteins were stained using SYPRO Ruby protein gel stain (ThermoFisher Scientific) as per the manufacturer’s instructions and visualized under UV light. Molecular weights of separated proteins were determined by relative mobility (Rf) analysis.

Protein bands which could be visualized by eye were excised and de-stained in a solution of 50% acetonitrile (ACN) in 50 mM ammonium bicarbonate (ABC) overnight at 37 °C with shaking. Gel slices were treated with 10 mM DTT at 60 °C for 30 min, and then alkylated in a 30 mM acrylamide solution for 30 min at room temperature, protected from light. The gel slices were then washed twice with 50 mM ABC and dehydrated with 100% ACN. Proteins were digested with 80 ng of trypsin (New England BioLabs) in 50 mM ABC at 37 °C overnight. Peptides were extracted by sonication of the gel slices in a solution of 50% ACN with 0.1% formic acid (FA).

For in-solution digestions, purified virions were denatured in a solution of 8 M urea in 50 mM ABC and centrifuged at 14,000 g for 10 min to remove debris. Samples were reduced in 5 mM DTT at 56 °C and alkylated with 25 mM iodoacetamide for 30 min at room temperature. Excess iodoacetamide was quenched by another addition of DTT to a final concentration of 5 mM. Samples were diluted in 50 mM ABC to bring the concentration of urea below 2 M. Approximately 12 µg of protein was incubated with 120 ng trypsin (1:100) or 60 ng chymotrypsin (1:200) overnight at 37 °C.

Ziptip (Millipore) cleanup was performed on all samples to remove impurities. Peptides were eluted in 0.1% FA and run on both the Orbitrap Elite and Triple Tof 5000 mass spectrometers. Data was analyzed with ProteinPilot software (SCIEX) using databases containing *Ae. albopictus* proteins and ABV polyprotein and RdRP amino acid sequences.

Identified peptides mapping to the ABV polyprotein with >50% confidence were further assigned to individual proteins (pVP2, VP3, VP4) manually in Excel based on the predicted sequence for each protein.

Mass spectrometry analysis for glycopeptide analysis is described in detail in Appendix B.

## 3. Results

### 3.1. Detection of Two Isolates of a Novel Virus with Distinctive dsRNA Immunostaining

During routine mosquito collections for arbovirus surveillance in NSW, Australia, 45 mosquito pools were identified as positive for unidentified viruses via detection of dsRNA in inoculated C6/36 cells by ELISA, using the MAVRIC system. The pools were further analyzed by a combination of fixed-cell ELISA using virus-specific mAbs and RT-PCR using virus-specific primers to detect selected ISVs known to circulate in Australian mosquito populations (Appendix A). This analysis identified 14 isolates of Liao ning virus (LNV), including one pool co-infected with *Alphamesonivirus* 1 (Nam Dinh virus, NDiV), which was first reported in Prow et al. [28] (Figure 1a). A subset of the remaining pools, which were negative for the tested ISVs, were selected for further analysis with MAVRIC in IFA. Two pools of *Ae. notoscriptus* mosquitoes collected in Ballina, NSW (177833 and 178287) shared a unique punctate, cytoplasmic dsRNA-staining pattern (Figure 1a,b). Next-generation sequencing of the two pools identified a birnavirus-like sequence with 37.52% identity to the RdRP of Rotifer birnavirus (RBV) by Blastx analysis (e-value 4e-138). This virus was tentatively named Aedes birnavirus (ABV). Comparison of the sequences from both pools showed high similarity at the nucleotide level (99.95% Segment A, 99.93% Segment B), suggesting that isolates 177833 and 178287 were likely of the same virus strain. No overt cytopathic effect (CPE) was observed in C6/36 cells infected with either isolate of ABV, while cells inoculated with the *Ae. notoscriptus* pool containing LNV and NDiV (179853) exhibited clearance of the monolayer (Figure 1c). No evidence of co-infection with other known viruses was observed for either pool containing ABV. Using ABV-specific primers (Appendix A), retrospective RT-PCR screening was performed on the 45 MAVRIC-positive unknown pools from NSW, along with selected *Aedes* mosquito pools from the Northern Territory and Western Australia, which were found to be MAVRIC positive but negative for all tested ISVs. However, no further isolates of ABV were identified (Figure 1a, Appendix A).

### 3.2. Phylogenetic Analysis

The segment A of ABV possesses 45.6% GC content and does not contain the slippery UUUUUUAA motif which is conserved among entomobirnaviruses [17,18,20]. Segment A gives rise to a polyprotein of 1239 amino acids, larger than the typical range of 972–1114 amino acids reported in other birnavirus polyproteins. Segment B has a GC content of 43.9% and encodes a VP1 which is 904 amino acids in length. The VP1 of ABV does not contain the large C-terminal domain which is present in RBV and the entomobirnaviruses (Appendix A) [2].

Phylogenetic analyses of both the nucleotide and amino acid sequences of VP1 and the amino acid sequence of the polyprotein indicated that ABV forms a well-supported clade with RBV, indicating a possible distant taxonomic association with this virus (Figure 2a,b, Figure 3b). Nucleotide-based phylogenetic analyses of the polyprotein showed ABV clustered with the *Entomobirnavirus* genera. However, this position was not as strongly supported (bootstrap value of 54%) (Figure 3a).

### 3.3. Analysis of ABV Structural Proteins

Transmission electron microscopy (TEM) on purified ABV revealed icosahedral particles with average diameters ranging from 64 to 70 nm (Figure 4a). Inoculation of C6/36 cells with two independently purified virus preparations indicated that the purified particles were infectious (Figure 4b). SDS-PAGE analysis of the purified virions revealed 2 major bands at approximate molecular weights of 38 and 28 kDa which were determined by mass spectrometry to contain VP3 and VP4, respectively (Figure 4c). Less-prominent bands were identified at approximate molecular weights of 113 kDa corresponding to the RdRP and 15, 16 and 7.5 kDa likely to be peptides derived from the C-terminus of pVP2 (Figure 4c). A distinct band at the expected size range for the mature birnavirus VP2 (40–55 kDa) was not identified [4,6,16,38,39]. Instead, peptides mapping to ABV VP2 were only identified in bands running at apparent molecular weights of 150, 113 and 70 kDa (Figure 4c, Appendix A). While our mass spectrometry analyses resulted in high coverage for VP3 and VP4 (93.8% and 84.5% respectively), the coverage for pVP2 was much lower (31.9%) (Figure 4c). Further denaturation of virions in 3 M urea did not alter the banding pattern of ABV proteins in SDS-PAGE, suggesting that the larger-than-expected size for VP2 was not due to dimerization of the protein (Figure 4d).

### 3.4. ABV Proteins Demonstrate Trypsin Resistance

Due to the lower coverage of VP2 observed in mass spectrometry analysis, the predicted peptides generated from trypsin cleavage of the polypeptide were investigated with ExPASy PeptideCutter. This analysis showed a lack of cleavage sites for trypsin in the predicted pVP2 sequence of ABV which resulted in generation of large peptides (23–97 aa) (Appendix A) which would be unlikely to be detected by mass spectrometry. To confirm this, side-by-side digestions using trypsin and chymotrypsin were performed on whole virions. Mass spectrometry analysis of the resulting peptides showed that while digestion with chymotrypsin resulted in higher coverage of all ABV proteins when compared to trypsin, this difference was most striking for pVP2 with a 35.9% increase (trypsin, 34.9% to chymotrypsin, 70.8%) (Table 1, Figure 5). This indicated that trypsin was not optimal for proteolytic digestion of ABV pVP2. However, due to its low cleavage site fidelity, chymotrypsin was deemed an unsuitable alternative for mass spectrometry analysis [40].

### 3.5. Sequence Analysis of ABV Polyprotein

In order to identify potential cleavage sites in the ABV polyprotein, amino acid sequence alignment was performed with the polyprotein sequences of ESV and birnaviruses for which cleavage sites have been experimentally determined [16,41,42]. Based on known cleavage sites for these viruses, the C-terminus of the mature ABV capsid protein was predicted at amino acid 437, giving rise to a protein with an estimated molecular weight of 47.2 kDa (Figure 6). Analysis of the polypeptide revealed two insertions of 61 and 85 amino acids between VP2 and VP4 which were not conserved in the other birnaviruses. The resulting pVP2 protein had a predicted molecular weight of 72.6 kDa, consistent with the size range at which we detected pVP2 peptides in SDS-PAGE analysis (Figure 4). Furthermore, peptides mapping to these inserted regions were detected during mass spectrometry analysis performed on both gel slices after SDS-PAGE separation and purified virions (Figure 5, Appendix A). No strong evidence of N- or O-linked glycans on any ABV proteins could be found using mass spectrometry (Appendix A). Sequential proteolytic digestion of ABV virions by Glu-C and trypsin did not improve coverage of pVP2 (Appendix A).

### 3.6. Production of Monoclonal Antibodies to ABV

To aid in additional screening for, and characterization of, ABV, purified virions were used to immunize mice for the production of polyclonal antisera and mAbs. Mouse immune serum raised to ABV was tested in Western blot against virus culture supernatant prepared with minimal serum content, virus stock supernatant and two preparations of purified ABV under unreduced or boiled and reduced conditions (Figure 7a). This analysis showed VP3 (approximate molecular weight of 38 kDa) to be the immunodominant protein in all preparations. The anti-ABV serum also displayed reactivity to a 50–60 kDa protein in cell culture supernatants, which was not observed in either preparation of purified virions. This protein most likely represents a component of FBS which was present at low levels (< 1%) in concentrated virus supernatant (sample 1) and at 10% *v*/*v* in virus stock supernatant (sample 2) (Figure 7a). The anti-ABV mouse immune serum had a strong neutralizing effect on ABV, inhibiting replication at all tested dilutions (1:20 to 1:2480).

Two hybridoma fusions using the spleens from immunized mice resulted in a panel of 13 mAbs with specificity to ABV in 20% acetone fixed-cell ELISA. Nine of the 13 mAbs were found to bind to VP3 in Western blot, consistent with the binding profile observed for the polyclonal serum (Figure 7b). The remaining four mAbs did not react in Western blot and may bind conformational or SDS-sensitive epitopes. Two mAbs, A4G7 and A6F6, were shown to have neutralizing capabilities in microneutralization assays and, based on this property, are predicted to bind the capsid protein VP2 (Figure 7b). Neither A4G7 nor A6F6 reacted in Western blot. None of the mAbs which reacted to VP3 in Western blot showed any neutralizing activity. As acetone fixative buffer is not optimal for inactivation of non-enveloped viruses, the anti-ABV mAbs were tested for use with formaldehyde fixation. This testing found 11 of the 13 mAbs were reactive against formaldehyde-fixed epitopes in fixed-cell ELISA (Figure 7b).

### 3.7. ABV Does Not Replicate in Vertebrate Cells In Vitro

ABV was tested for replication in a panel of vertebrate cell lines derived from mammalian, avian, amphibian and reptilian origins. At 5 dpi, no viable virus could be recovered from supernatants harvested from SW13, BSR, A459, IFNAR-/- MEFs, MDCK or VSW cells (Table 2, Appendix A). Low levels of virus were detected in DF-1, Vero and A6 cell supernatant at 5 dpi. However, viral titers in these supernatants were lower than that of supernatants harvested at 2 hpi, suggesting that this represented residual virus of the inoculum rather than productive replication. Supporting this, no dsRNA could be identified in any of the vertebrate cell lines by IFA at 5 dpi (Appendix A). This suggests ABV possesses an insect-specific phenotype and reflects the findings for other birnaviruses isolated from mosquitoes.

### 3.8. ABV Displays Aedes-Specific Tropism In Vitro

While some ISVs are highly specific for a single mosquito species or genera [25,30,43], others show the ability to infect a broad range of mosquito species [27,28,29,33]. Therefore, ABV was assessed for replication in cell lines derived from *Aedes*, *Culex* and *Anopheles* mosquitoes. IFA revealed cytoplasmic dsRNA staining in RML-12 cells (*Ae. albopictus*, RNAi-competent) like that observed in C6/36 cells (*Ae. albopictus*, RNAi-deficient) (Figure 8a). In contrast, infected Chao Ball (*Cx. tarsalis*) and Mos55 (*An. gambiae*) cells were characterized by a single focus of perinuclear dsRNA staining (Figure 8a, inset). ABV replicated in RML-12 cells to an average titer of 10^5.7^/mL by 7 dpi, approximately 1 log lower than the mean titer reached in C6/36 cells (Figure 8b). Comparatively, titers of ABV in supernatant harvested from Chao Ball cells at 7 dpi were unchanged from those detected at 2 hpi with mean titers for both time points at 10^1.8^/mL. Similarly, only a slight, albeit significant (*p* = 0.017, t test), increase in titer was detected in Mos55 cells at 7 dpi (mean titer: 10^2.08^/mL) compared to 2 hpi (mean titer: 10^1.65^/mL) (Figure 8b). To investigate the unique fluorescence pattern observed in Figure 8a, ABV-infected Chao Ball, Mos55 and C6/36 cells were fixed in 100% acetone or formaldehyde fixative buffer and immunolabeled for dsRNA. Under formaldehyde-fixed conditions, it was found that the punctate signal in Chao Ball and Mos55 cells was emphasized, whilst fixation with acetone resulted in a more diffuse cytoplasmic signal (Figure 8c). Finally, immunolabeling with ABV-specific mAbs suggested that both VP2 and VP3 were produced in ABV-infected Chao Ball and Mos55 cells, suggesting that some viral replication occurs in these cell lines (Figure 8d).

## 4. Discussion

In this study, we describe the first isolation and characterization of a unique birnavirus from Australian mosquitoes. This virus was tentatively named Aedes birnavirus (ABV) after the *Ae. notoscriptus* mosquito from which it was isolated. Phylogenetic analyses indicate that ABV is not likely to be a member of the *Entomobirnavirus* genus. Instead, ABV forms a clade with Rotifer birnavirus (RBV), an unassigned member of the *Birnaviridae* that infects the microscopic aquatic animal *B. plicatilis* [4]. ABV was identified from a subset of mosquito pools collected in NSW, Australia, which were found to be positive for long (> 30 bp) double-stranded RNA (dsRNA) in fixed-cell ELISA using the MAVRIC system [26]. While many ISVs are reported as co-isolations [27,28,30], RT-PCR screening and deep sequencing performed on supernatant from the two mosquito pools containing ABV did not indicate the presence of any other virus species. A total of 77 MAVRIC-positive mosquito pools were tested for ABV, but no further isolates have been identified. It should be noted that retrospective screening in this study focused primarily on *Aedes* mosquito pools from Western Australia and the Northern Territory and excluded pools already known to contain other ISVs. In future studies it would be of interest to screen mosquito pools from NSW and south-east Queensland to account for a narrow geographic range which has been reported for some ISVs [30,33,43].

Purification of ABV by CsCl gradient yielded virions of icosahedral appearance with approximate diameters of 64–70 nm, within the reported range for other birnaviruses (55–70 nm) [15,16,20]. SDS-PAGE and mass spectrometry analysis of these particles confirmed the presence of VP1, VP3, VP4, and the small structural peptides, consistent with reports for other birnaviruses [16,39]. While bioinformatic analyses predicted the mature VP2 protein to be approximately 47.2 kDa in size, VP2 peptides could only be detected in bands in SDS-PAGE between 70 and 150 kDa. Analysis of the polyprotein sequence of ABV revealed two unique insertions between the pVP2 and VP4 genes which are likely to be incorporated in the C-terminus of pVP2. The presence of these inserted sequences in the ABV virion was confirmed by detection of peptides mapping to these regions by mass spectrometry. With these insertions, the resulting pVP2 protein has an estimated molecular weight of 72.6 kDa, consistent with the size at which this protein was detected in our analyses. Thus, it appears that these purified virions predominantly contain the precursor form of the capsid protein. Studies on infectious pancreatic necrosis virus (IPNV) and infectious bursal disease virus (IBDV) have provided a model for birnavirus virion assembly and maturation. In this model, the immature or “provirion” predominantly contains pVP2 and has a diameter which is slightly larger than the mature form [15,44]. In the case of IPNV, immature virions were isolated by lysis of infected cells, while mature particles were isolated from cell supernatant [15]. Maturation of the virion occurs via successive cleavages at the c-terminal end of pVP2 resulting in the mature form of VP2 and 3–4 C-terminal-derived peptides which remain associated with the virion [3]. Our data suggests that particles recovered from ABV infected cells contain both pVP2 and a number of small peptides which may be derived from the C-terminus of pVP2. At this stage we are unable to confirm whether these particles represent the immature or mature form of ABV. However, we have shown that these particles are infectious and as ABV does not cause overt cytopathic effect, it appears that this virion form is released from infected cells via a non-lytic pathway.

Our mass spectrometry analyses suggested that ABV proteins, particularly pVP2, are suboptimally cleaved by trypsin even when preceded by Glu-C digestion. Limited information appears to be available on the protease resistance of other birnaviruses, but both trypsin and Lys-C digestion have been used for mass spectrometry analysis of birnavirus proteins by others [16,44].

In this study, we report the first production of immune sera and mAbs to a birnavirus isolated from mosquitoes. Analysis of these antibodies by Western blot suggested that VP3 is the immunodominant protein. However, both the immune sera and two mAbs, which did not react in Western blot, showed neutralizing activity against the virus, indicating that the predominant VP2 antibody response was to conformational or SDS-sensitive epitopes of the protein, consistent with reports for IPNV and IBDV [45,46].

No evidence of ABV replication was detected in an extensive panel of vertebrate cell lines, suggesting that the virus may possess an insect-specific phenotype similar to what has been reported for the entomobirnaviruses [16,17,20]. However, considering the relatedness of this virus to RBV, and the presence of aquabirnaviruses in Australia [47], it would be of interest to investigate the potential of cell lines derived from fish and other aquatic species to support ABV replication.

Intriguingly, Thirlmere virus, a birna-like virus for which no sequence data is currently available has been isolated from lake water [48]. This virus was found to be antigenically related to Drosophila X virus and replicated in *D. melanogaster* cells, suggesting that a birnavirus of diptera may be present in aquatic environments [48].

While ABV was able to infect and replicate to similar titers in two cell lines derived from *Ae. albopictus* mosquitoes, only limited replication was observed in Mos55 (*An. gambiae*) and Chao Ball (*Cx. tarsalis*) cells. Under formaldehyde-fixed conditions, ABV-infected Mos55 and Chao Ball cells were characterized by a single punctate focus of dsRNA staining that contrasted with the diffuse-cytoplasmic immunostaining observed in *Aedes*-derived cell lines. However, when cells were fixed in acetone, dsRNA immunostaining became more diffuse, suggesting that the localization of staining may be influenced by the type of fixative used. A recent paper has demonstrated that use of alcohol and formalin-based fixatives can have a dramatic effect on the cellular localization of proteins [49].

Our in vitro data may indicate an *Aedes*-specific tropism for ABV, as has been described for another Australian ISV, Parramatta River virus [30]. This result is particularly interesting since Mosquito X virus (MXV) and Culex Y virus (CYV), two highly genetically similar entomobirnaviruses, were detected in mosquitoes of different genera (MXV in *An. sinensis*; CYV in *Cx. pipiens*) [17,18]. Furthermore, CYV infection has been reported in various cell lines derived from *Culex*, *Aedes* and *Drosophila* origins [50,51]

Our characterization of ABV suggests that it is a unique member of the *Birnaviridae* family. To our knowledge, we also report the first panel of mAbs raised to a birnavirus derived from mosquitoes. These mAbs will assist in future studies on the ecological niche of this virus and its relatedness to other members of the *Birnaviridae*. The discovery of ABV further adds to our knowledge of the virome of mosquitoes within Australia.

## Figures and Tables

**Figure 1 viruses-12-01056-f001:**
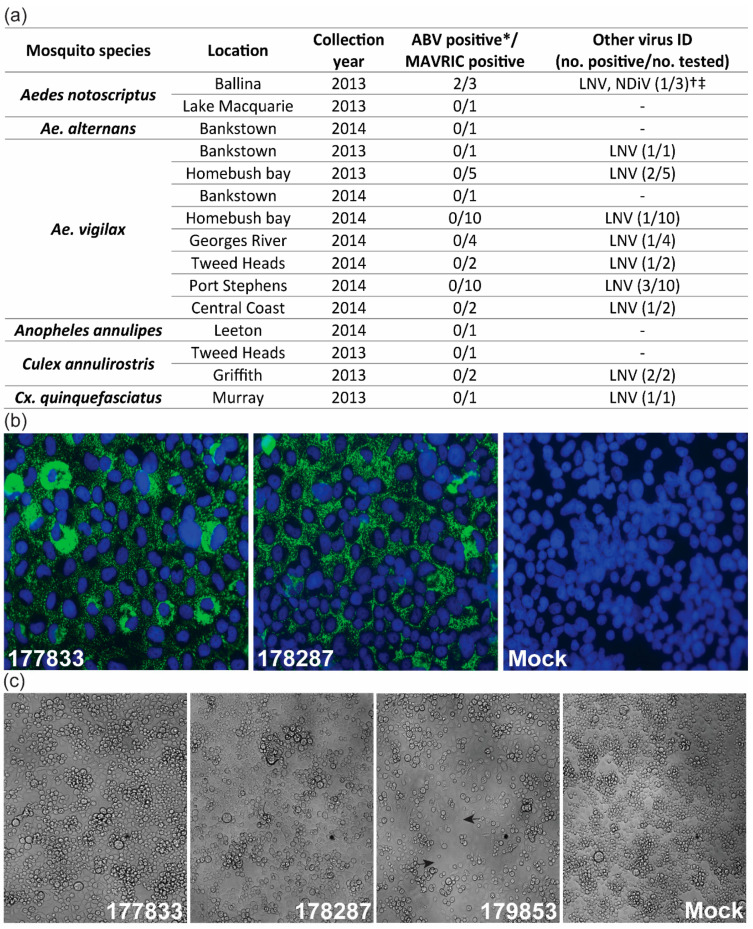
Isolation and prevalence of Aedes birnavirus (ABV). (**a**) Summary of screening for ABV and known insect-specific viruses (ISVs) in MAVRIC-positive mosquito pools collected in NSW between 2013 and 2014. (**b**) Anti-double-stranded RNA (dsRNA) immunolabeling in C6/36 cells inoculated with mosquito pools 177833 and 178287, or mock infected (media only) in immunofluorescence assay. Images were taken at 63× magnification. Green: dsRNA. Blue: nuclei. (**c**) C6/36 cells infected with ABV isolates 177833 and 178287, NDiV/LNV co-infected isolate 179853 and mock-infected cells. Gaps in the monolayer typical of NDiV/LNV infection are depicted by arrows for isolate 179853. Cell monolayer images were taken at 20x magnification. * Screened by RT-PCR with ABV-specific primers. The † LNV/NDiV-positive pool did not contain ABV. ‡ Ballina LNV/NDiV isolate originally reported in Prow et al. [28].

**Figure 2 viruses-12-01056-f002:**
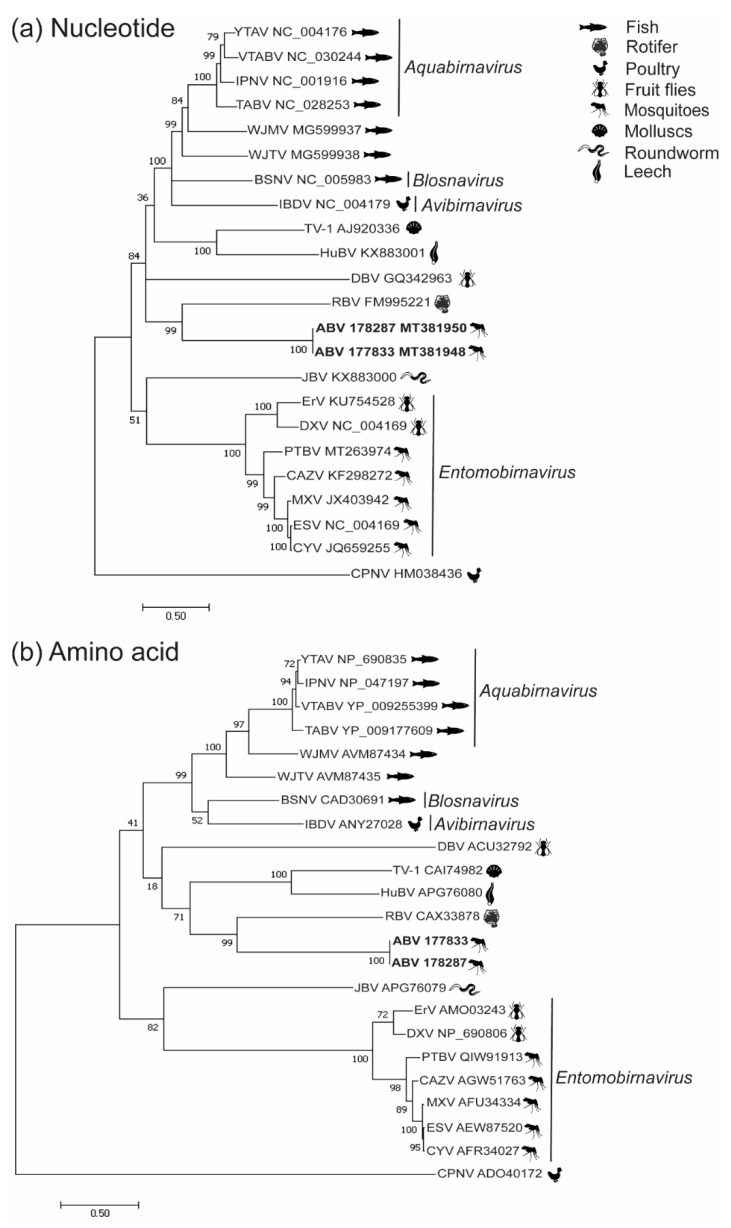
Phylogenetic analysis of ABV VP1. Unrooted maximum likelihood trees constructed from (**a**) VP1 nucleotide coding sequences (CDS) aligned using the MUSCLE algorithm and (**b**) VP1 amino acid sequences aligned with MAFFT. The broad host-species classification for each virus is indicated by icons next to the virus name. Currently recognized genera are indicated in italics. Maximum likelihood analyses were performed in MEGA 7.0.26 using the general time-reversible (GTR) model with gamma distribution (5 categories ( + G, parameter = 1.8479)) and invariable variation for some sites ([ + I], 2.92% sites) for nucleotide sequences and the Jones–Taylor–Thornton (JTT) model with gamma substitutions and no deletions for amino acid sequences.

**Figure 3 viruses-12-01056-f003:**
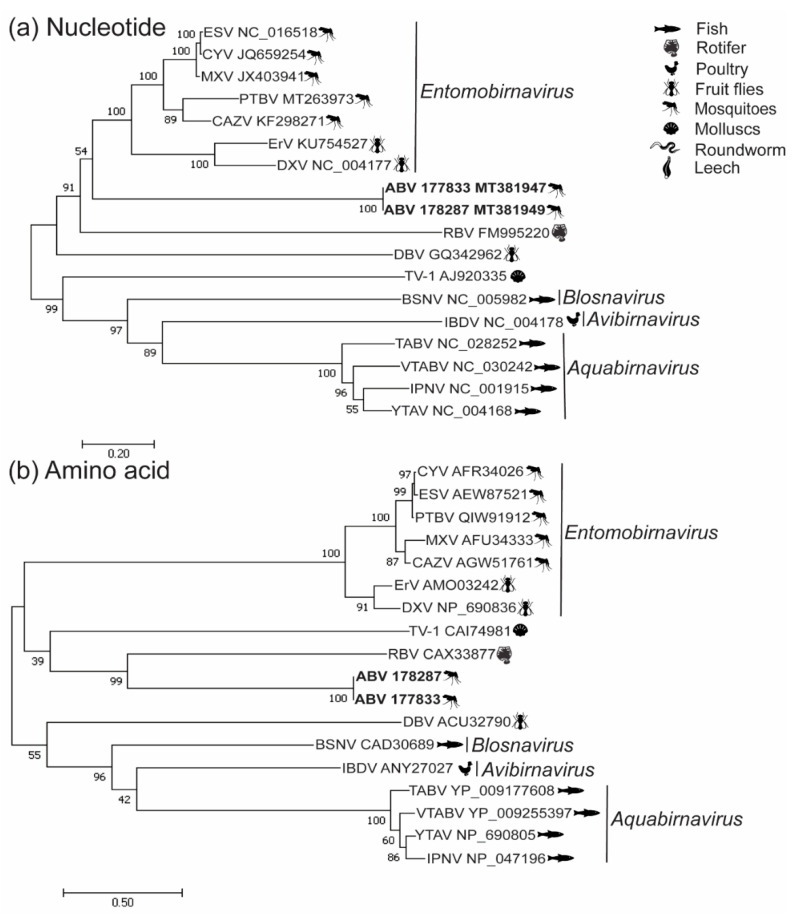
Phylogenetic analysis of ABV polyprotein. Unrooted maximum likelihood trees constructed from (**a**) nucleotide coding sequences (CDS) for birnavirus polyproteins aligned using the MUSCLE algorithm and (**b**) polyprotein amino acid sequences aligned with MAFFT. The broad host-species classification for each virus is by icons next to the virus name. Currently recognized genera are indicated in italics. Maximum likelihood analyses were performed in MEGA 7.0.26 using the general time-reversible (GTR) model with gamma distribution (5 categories ( + G, parameter = 1.8479)) and invariable variation for some sites ([ + I], 2.92% sites) for nucleotide sequences and the Jones–Taylor–Thornton (JTT) model with gamma substitutions and no deletions for amino acid sequences.

**Figure 4 viruses-12-01056-f004:**
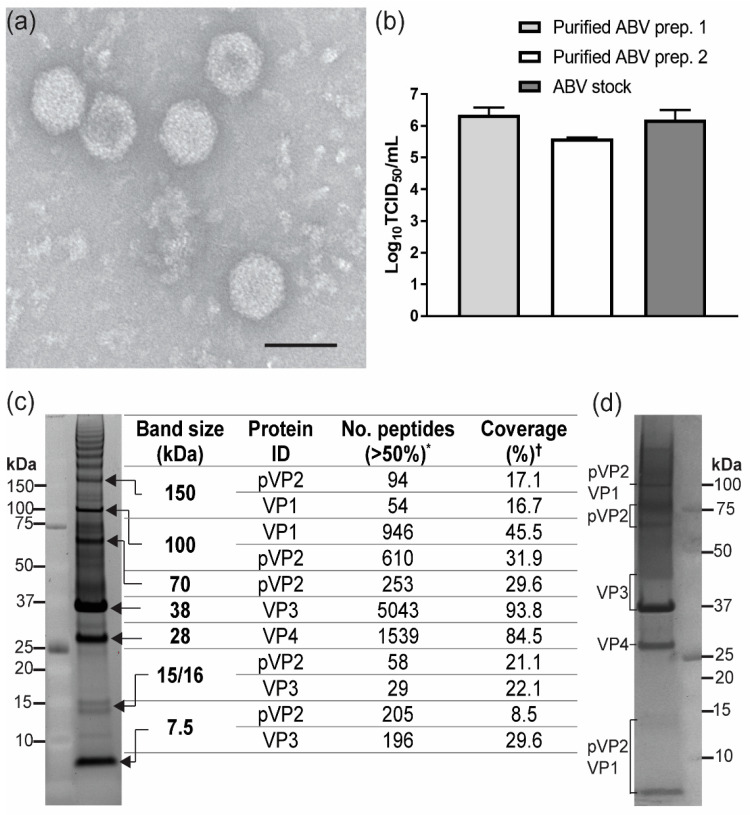
Protein analysis of ABV. (**a**) Transmission electron micrograph of purified ABV particles negatively stained with uranyl acetate and imaged at 200 K. Scale bar represents 100 nm. (**b**) Infectivity of two preparations of CsCl gradient-purified ABV particles in C6/36 cells as compared to ABV-infected cell supernatant used at a known MOI (0.1). Values graphed are averaged virus titers in supernatants taken from infected cells at 7 days post-infection (dpi). Error bars represent standard deviation in titers between 3 replicate wells. (**c**) Ruby stained image of SDS-PAGE analysis on purified ABV boiled and reduced with 1 M DTT, and results of mass spectrometry identification of excised protein bands as indicated by arrows. Protein ID refers to top ranking proteins identified in each band by mass spectrometry analysis. * No. peptides, number of peptides identified with equal to or greater than 50% confidence; ^†^ Coverage, percentage of protein covered by peptides identified with >50% confidence. (**d**) Ruby stained image of SDS-PAGE analysis of ABV further denatured in 3 M urea. Protein identities based on mass spectrometry are indicated next to each band. Kaleidoscope protein ladder was run alongside all samples.

**Figure 5 viruses-12-01056-f005:**
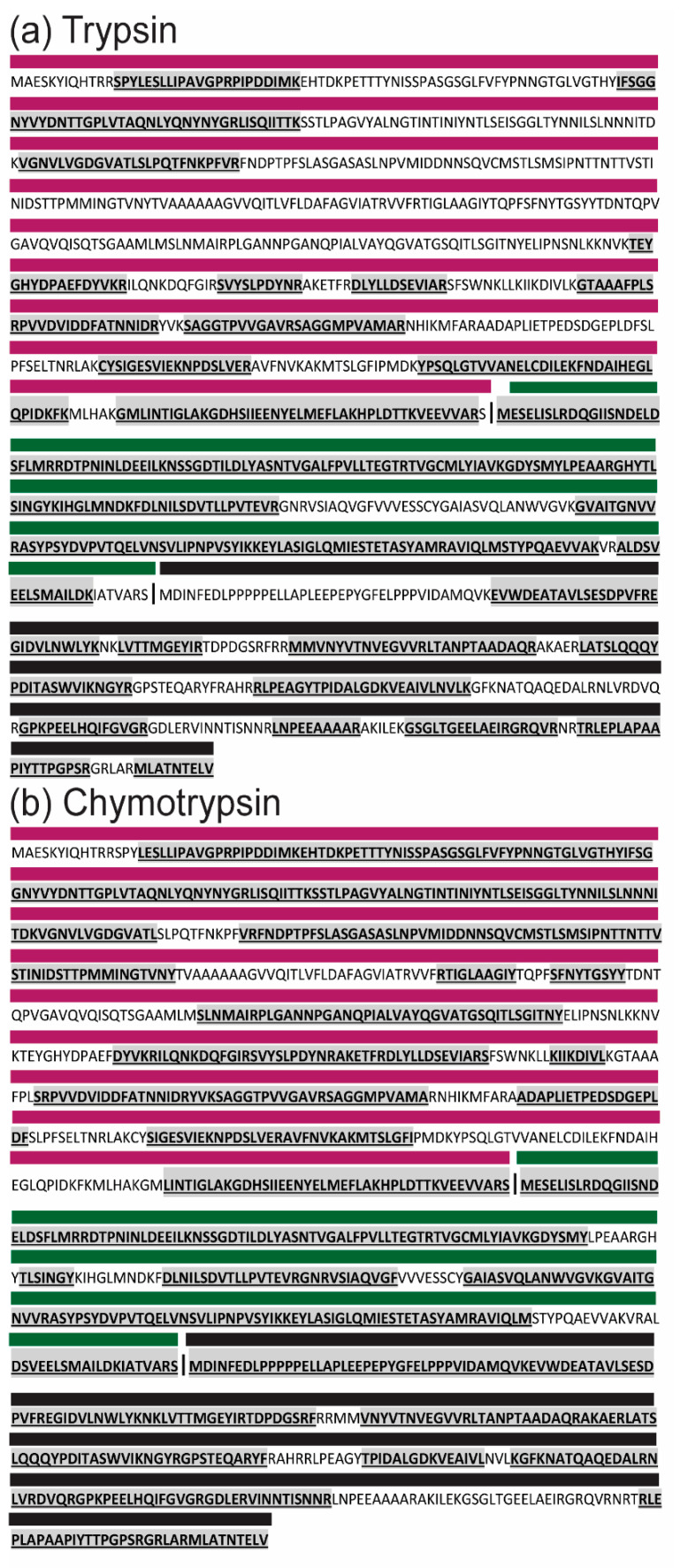
Digestion of ABV proteins by trypsin and chymotrypsin. Schematic showing ABV polyprotein sequence identified by mass spectrometry following digestion with (**a**) trypsin or (**b**) chymotrypsin. Sequence detected by mass spectrometry is highlighted in grey and underlined. Colored bars above sequence denote predicted individual proteins: pVP2, magenta; VP4, green; VP3, black.

**Figure 6 viruses-12-01056-f006:**
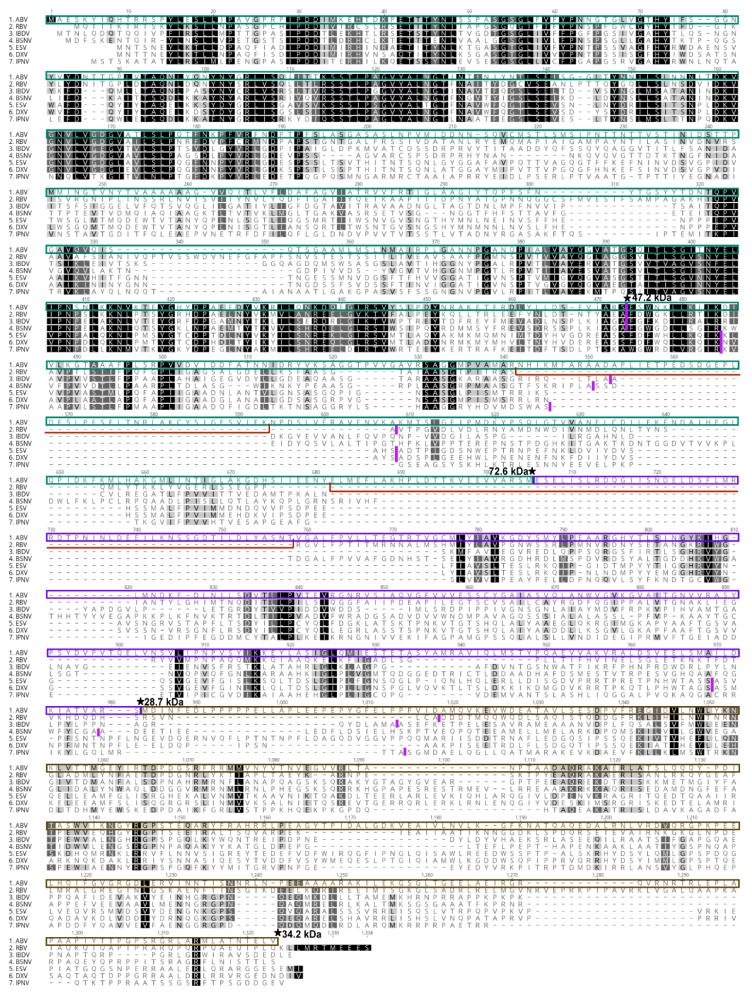
Clustal W alignment of amino acid polyprotein sequences for ABV, RBV (CAX33877), IBDV (ANY27027), BSNV (CAD30689), ESV (AEW87521), DXV (NP_690836) and IPNV (NP_047196). Non-conserved insertions of 61 and 85 amino acids in the ABV polyprotein are indicated by red lines. Vertical lines in magenta depict published polyprotein cleavage sites for these birnaviruses [41,42]. ★ Denotes predicted cleavage sites for ABV based on mass spectrometry data; Molecular weights predicted using ExPASy compute pI/MW tool are marked at the c-terminus of each resulting protein. Green box, pVP2. Purple box, VP4; brown box, VP3.

**Figure 7 viruses-12-01056-f007:**
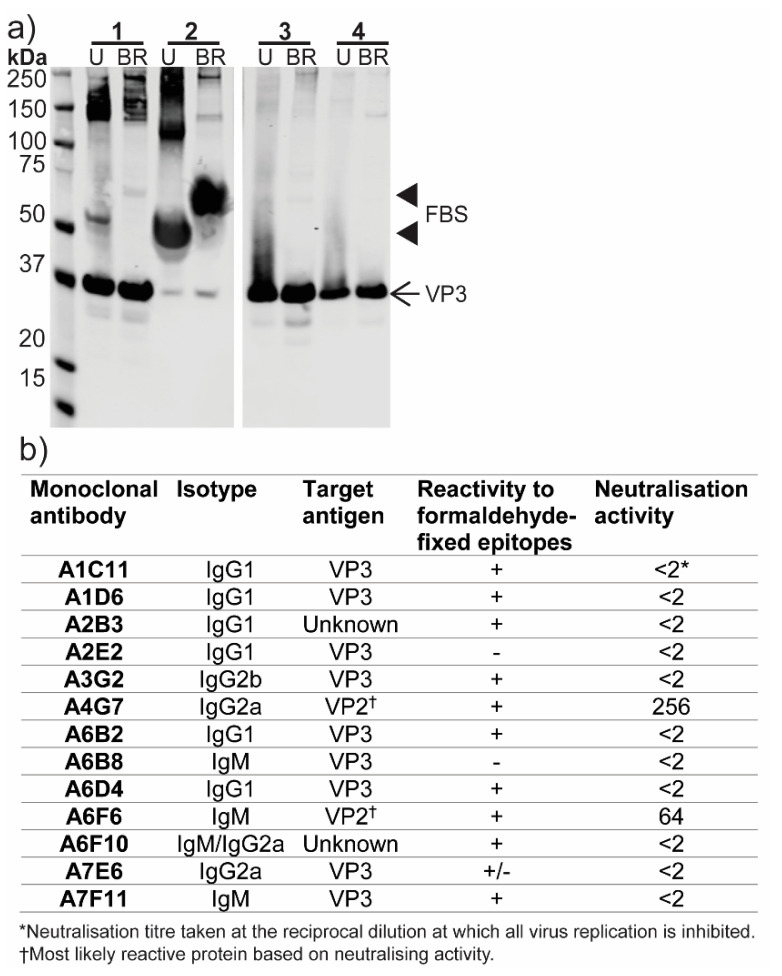
Antibody response to ABV. (**a**) Reactivity of anti-ABV mouse immune serum to ABV in Western blot. Lane 1, supernatant from ABV-infected C6/36 cells grown in minimal (< 1%) FBS; 2, ABV virus stock supernatant from C6/36 cells, supplemented with 10% FBS; 3, CsCl gradient-purified preparation one of ABV in PBS; 4, CsCl gradient-purified preparation two of ABV in PBS. U, unreduced; BR, boiled and reduced. (**b**) Summary of ABV-specific monoclonal antibody panel. † Most likely target based on neutralizing activity. * Highest reciprocal dilution at which virus infectivity is inhibited. –, OD450 nm <0.25 and less than two times the average OD of the negative control; +, OD450 nm is >0.5 and at least two times greater than the average OD of negative control; +/-, positive result but inconsistent between replicates.

**Figure 8 viruses-12-01056-f008:**
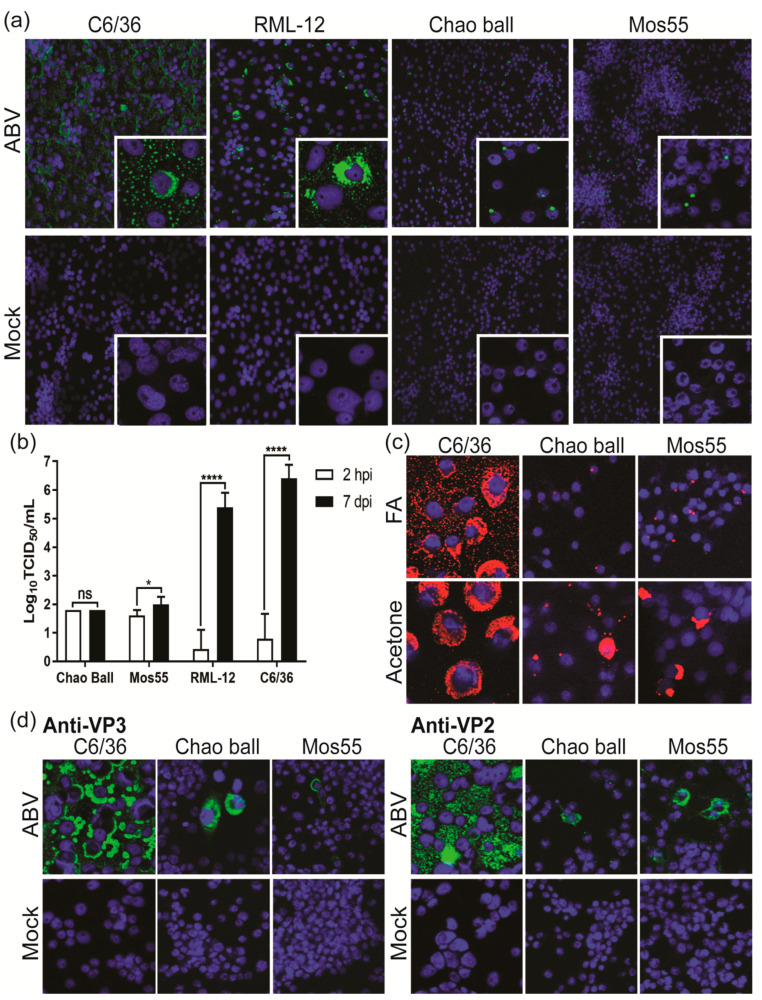
ABV replication in mosquito cell lines. (**a**) IFA performed on mosquito cells infected with ABV or mock infected (media only) fixed with 4% formaldehyde (FA) and 0.5% Triton X-100 at 7 dpi and immunolabeled for dsRNA (green) and nuclei (blue). Main panel images taken at 40x and inset at 63x magnifications. (**b**) Mean titers in supernatants harvested from mosquito cells at 2 hpi (white) and 7 dpi (black). Error bars represent standard deviations between three replicates. Statistical significance was calculated by t tests using the Holm–Sidak method, with alpha = 0.05. Each row was analyzed individually, without assuming a consistent SD. ****, *p* < 0.0001; *, *p* < 0.05; ns, not significant. (**c**) Localization of dsRNA (red) in ABV-infected mosquito cells fixed with either 100% acetone, or 4% formaldehyde + 0.5% Triton X-100. DsRNA (red); nuclei (blue). Images taken at 63x magnification. (**d**) ABV or mock-infected mosquito cells stained with VP3- and VP2-specific mAbs. ABV protein (green); nuclei (blue). Images taken at 40x magnification. C6/36, RNAi-deficient *Ae. albopictus*; RML-12, RNAi-competent *Ae. albopictus*; Chao Ball, *Culex tarsalis*; Mos55, *Anopheles gambiae*.

**Table 1 viruses-12-01056-t001:** Mass spectrometry analysis of peptides produced by digestion of ABV whole virions by trypsin or chymotrypsin.

Protein Identified	Trypsin		Chymotrypsin	
	No. Peptides (50%) *	% Coverage (50) ^†^	No. Peptides (50%) *	% Coverage (50) ^†^
VP1	161	38.4	145	43.04
Polyprotein	605	54.7	476	74.6
pVP2	197	34.9	250	70.8
VP3	237	60.6	108	82.08
VP4	171	83.7	109	84.09

* No. peptides (50%), number of distinct peptides with at least 50% confidence; ^†^ % coverage (50), percentage of matching amino acids from identified peptides with equal to or greater than 50% confidence divided by total no. amino acids in protein sequence.

**Table 2 viruses-12-01056-t002:** Summary of findings for ABV infection of vertebrate cells.

Cell Line	Cell Origin	ABV Replication	Mean Titer Recovered (/mL) ^†^	
			2 hpi	5 dpi
DF-1	Avian	-	10^1.8^	10^0.87^
Vero	Monkey	-	10^1.8^	10^1.41^
SW13	Human	-	10^1.3^	<10^1.3^ *
A549	Human	-	10^1.78^	<10^1.3^
MDCK	Canine	-	10^1.05^	<10^1.3^
BSR	Rodent	-	10^2.02^	<10^1.3^
MEF IFNAR^-/-^	Rodent	-	10^1.11^	<10^1.3^
VSW	Reptile	-	10^1.71^	<10^1.3^
A6	Amphibian	-	10^1.71^	10^1.45 #^
C6/36	Mosquito	+	10^2.13^	10^5.8^

^†^ Mean titer (TCID_50_/mL) averaged across three replicates per time point. * Limit of detection. ^#^ Virus detected in two of three replicates. -, ABV did not replicate; +, ABV replication; hpi, hours post-infection; dpi, days post-infection.

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
