# Peer review of "A Unique Relative of Rotifer Birnavirus Isolated from Australian Mosquitoes"

_viruses, 2020, doi:10.3390/v12091056_

Round 1

Reviewer 1 Report

This manuscript describes the isolation and characterization of a so-called Aedes birnavirus, a double-stranded RNA virus that infects mosquitoes in Australia and is believed to be related to the Rotifer birnavirus. The virus was isolated with the help of the monoclonal antibodies to viral RNA intermediates in cells (MAVRIC) system followed by deep sequencing. The virus was further characterized by using transmission electron microscopy and other in-vitro tests.

The method applied is not novel but virus discovered is new. The manuscript is well written but has the following minor shortcomings:

Too much reference to external work in materials and methods- makes it difficult to understand the manuscript without reading other papers. More brief descriptions of methods should be included so the manuscript can stand alone. An example is MAVRIC screening of samples (section 2.1).

Lines 123-5 de novo assembly – how exactly was this done? What was the length of the reads used? What parameters were used for de novo assembly?

In line 302, “clearance of the monolayer in the pool containing LNV and NDV (179853)”, this is not obvious to me. Please insert an arrow in the figure to show this. The virus does not induce CPE, what would be its clinical significance?

Line 225- 230. what concentration of formaldehyde was used? 4%?

Author Response

We are grateful to the reviewer for their insightful comments and have detailed how we have addressed each comment in blue text.

1. The method applied is not novel but virus discovered is new. The manuscript is well written but has the following minor shortcomings:

Author response: the description of the MAVRIC system as "novel" has been changed to "broad-spectrum" in the abstract.

2. Too much reference to external work in materials and methods- makes it difficult to understand the manuscript without reading other papers. More brief descriptions of methods should be included so the manuscript can stand alone. An example is MAVRIC screening of samples (section 2.1).

Author response: Additional description of the methods has been added and reference to other papers have been removed at the following sections:

  1. Lines 93 – 123: a description of the protocol for MAVRIC screening and fixed cell ELISAs has been included.
  2. Lines 256 – 265: a description of the protocol for immunofluorescence assay has been included
  3. Lines 278 – 287: a description of the protocol for hybridoma fusion has been included.

3. Lines 123-5 de novo assembly – how exactly was this done? What was the length of the reads used? What parameters were used for de novo assembly?

Author response: We have included additional details for the de novo assembly method at lines 148 – 151.

4. In line 302, “clearance of the monolayer in the pool containing LNV and NDV (179853)”, this is not obvious to me. Please insert an arrow in the figure to show this. The virus does not induce CPE, what would be its clinical significance?

Author response: Arrows indicating areas of monolayer clearance have been inserted in Figure 1c as requested. In addition, the contrast has been increased on the images and the images have been rotated and enlarged slightly for improved visibility. The original and contrast-edited images have been uploaded for transparency.

We are unable to comment on the clinical significance of ABV-infection in mosquitoes without in vivo assessment. In addition, ABV did not replicate in any vertebrate cell lines tested.

5. Line 225- 230. what concentration of formaldehyde was used? 4%?

Author response: we have now added “described in section 2.1” after formaldehyde fixative buffer at the sentence in question. In section 2.1 we describe the contents of formaldehyde fixative buffer.

Reviewer 2 Report

This is an extremely well written study that thoroughly explores the characterization of a novel virus from Aedes mosquitoes in Australia.

General Comments:

The comparison of trypsin and chymotrypsin digestion of the viral proteins is specific to ABV and not pertinent to comparison to other viruses. Therefore, the technical treatment of this comparison could be moved to Supplemental Data or left out entirely. 

Specific Comments:

lines 30, 53, 556-minor spelling error

Line 52-62. This paragraph is well-written and is appropriate for a thesis dissertation, but is not relevant to the topic of this research paper, and can be deleted.

Lines 96-98. Mosquitoes were collected in New South Wales. Here you refer to Western Australia and Darwin. For the reader not geographically familiar with Australia, please connect the dots.

Line 236 xylazil? Same as xylazine?

Lines 252 and 253 "by western blot" rather than "in western blot"

Line 417 formatting issue.

Line 491 Table-formatting issue

Line 582 missing comma

Author Response

We thank the reviewer for their thorough and detailed review of the manuscript. Please find our responses to the comments in blue text below.

1. The comparison of trypsin and chymotrypsin digestion of the viral proteins is specific to ABV and not pertinent to comparison to other viruses. Therefore, the technical treatment of this comparison could be moved to Supplemental Data or left out entirely. 

Author response: We would like to request that this data remain in the main text of the paper as the comparison of trypsin and chymotrypsin digestion provides context for our characterization of the viral proteins by mass spectrometry. In addition, we feel that this data will provide valuable information for readers using mass spectrometry to characterise novel birnaviruses.

2.. Comment 1: lines 30, 53, 556-minor spelling error –

Author response:

On line 30 ‘notosciptus’ has been corrected to ‘notoscriptus

On line 53 ‘binavirus’ has been corrected to ‘birnavirus’

On line 556 ‘birnvairus’ has been corrected to ‘birnavirus’

3. Line 52-62. This paragraph is well-written and is appropriate for a thesis dissertation, but is not relevant to the topic of this research paper, and can be deleted.

Author response: the paragraph has been cut down to three sentences briefly describing the maturation process of birnaviruses which we believe is relevant to our characterization of the ABV virion.

4. Lines 96-98. Mosquitoes were collected in New South Wales. Here you refer to Western Australia and Darwin. For the reader not geographically familiar with Australia, please connect the dots.

Author response: A supplementary figure has been created (Figure S1) containing a map of Australia depicting the regions of mosquito collection sites and the additional mosquito pool details which were originally included in Table S2.

5. Line 236 xylazil? Same as xylazine?

Author response: ‘xylazil’ has been changed to ‘xylazine’

6. Lines 252 and 253 "by western blot" rather than "in western blot"

Author response: “by western blot” has been changed to “in western blot” on both lines.

7. Line 417 formatting issue.

Author response: Formatting issue has been corrected by deleting the word “sequence” and wording has been changed from “in sequence” to “in protein sequence”

8.Line 491 Table-formatting issue

Author response: the formatting issue has been corrected by removing the border below “DF-1” and “avian” and unbolding the two words.

9. Line 582 missing comma

Author response: Line 630 - A comma has been added after “Thirlmere virus”